# Smart Work Injury Management (SWIM) System: A Machine Learning Approach for the Prediction of Sick Leave and Rehabilitation Plan

**DOI:** 10.3390/bioengineering10020172

**Published:** 2023-01-28

**Authors:** Peter H. F. Ng, Peter Q. Chen, Zackary P. T. Sin, Sun H. S. Lai, Andy S. K. Cheng

**Affiliations:** 1Department of Rehabilitation Science, The Hong Kong Polytechnic University, Hong Kong, China; 2Department of Computing, The Hong Kong Polytechnic University, Hong Kong, China; 3Total Rehabilitation Management (HK) Limited, Hong Kong, China

**Keywords:** work injury, rehabilitation plan, rehabilitation case management, artificial intelligence, variational autoencoder, interactive dashboard, electronic health record

## Abstract

As occupational rehabilitation services are part of the public medical and health services in Hong Kong, work-injured workers are treated along with other patients and are not considered a high priority for occupational rehabilitation services. The idea of a work trial arrangement in the private market occurred to meet the need for a more coordinated occupational rehabilitation practice. However, there is no clear service standard in private occupational rehabilitation services nor concrete suggestions on how to offer rehabilitation plans to injured workers. Electronic Health Records (EHRs) data can provide a foundation for developing a model to improve this situation. This project aims at using a machine-learning-based approach to enhance the traditional prediction of disability duration and rehabilitation plans for work-related injury and illness. To help patients and therapists to understand the machine learning result, we also developed an interactive dashboard to visualize machine learning results. The outcome is promising. Using the variational autoencoder, our system performed better in predicting disability duration. We have around 30% improvement compared with the human prediction error. We also proposed further development to construct a better system to manage the work injury case.

## 1. Introduction

More than 90% of injured workers in Hong Kong receive occupational rehabilitation services in hospitals or rehabilitation centres, operated exclusively by the Hong Kong Hospital Authority. As occupational rehabilitation services are part of the public medical and health services offered to the entire population in the territory, injured workers are placed on a waiting list for occupational rehabilitation services along with other patients. Long waiting times for treatments and services can often lead to workers missing the “golden period” for rehabilitation, resulting in delays in recovery and returning to work. After setting up the services model in the private market, we foresee that more injured workers will be willing to receive private occupational rehabilitation services and the job bank for return-to-work (RTW) decisions. To facilitate an effective and efficient service in the private sector, we need a trustable and explainable system for predicting the sick leave and rehabilitation plan, especially for junior case managers. We used 90,154 work injury records and developed our smart work injury management (SWIM) system, which provides prediction using multi-dimensional data and machine learning approaches. Compared with predictions by case managers, we provide a better prediction result. We also developed an interactive dashboard to visualize the machine learning result and embedded it into the routine work of case managers. It offers an explainable result of what happens from the date of the accident to the close date to the case managers. 

## 2. Literature Review

The massive amount of electronic health records (EHRs) data has become a rich resource for research. EHRs data refer to collecting patient-related data, including patient information, hospital information, and diagnosis information. Although the primary use of EHRs data in the clinical system is to help doctors access patients’ previous records quickly and improve efficiency, researchers are now interested in discovering and revealing potentially valuable patterns from EHRs data. The early analyses of EHRs data relied on traditional statistical techniques. However, with the development of computer hardware, analysing and storing a massive amount of data has become possible and prevalent. Using up-to-date techniques such as machine learning could reveal many possible truths that have not been discovered before. EHRs data have been used for patient clustering [1], disease prediction, suicide prediction, drug–drug interaction [2], Chinese herb prescription [3] and clinical decision support [4]. Moreover, researchers have tried to adapt advanced techniques to analyse the EHRs data in recent years, such as using deep learning techniques to recognize doctors’ handwriting, speech recognition, and natural language processing [5]. Because of the importance of EHRs data, part of the research focuses on analysing massive datasets to produce feasible treatment recommendations [5,6]. The others might focus on improving EHRs data from different aspects, such as data storage [7,8,9] and data types [10].

Among that research, using EHRs data to predict patient trajectory with data-driven clinical support is highlighted. Some researchers propose an efficient way for knowledge representation of EHRs data and further developed a recommendation system to support doctors [11,12,13,14]. Alternatively, some even use a purely mathematical way to calculate a future medical event [15]. Others might focus on the machine learning approach, which can predict in several fields [16,17].

Many pieces of research focus on knowledge representation of clinical data to construct the data to easily search for similar cases. Some methodology would focus more on temporal data, and they would extract the timestamp from EHRs text and form a day-record vector to highlight the importance of temporal information [16]. Recently, knowledge representation research work also pays attention to temporal information, such as constructing temporally aware vectors from timestamped text data [18]. Spiotta further uses the answer set programming method to point out that temporal discrepancies between clinical guidelines and rudimentary medical knowledge are possibly due to prioritized execution [4]. 

As machine learning techniques are developing quickly today, these have also become a hot topic in clinical research [5,6]. Rajkomar used a deep learning model to achieve high accuracy for predicting events such as in-hospital mortality [19], 30-day unplanned readmission and prolonged length of stay [15]. Rahimian compared the random forest model and gradient boosting classifier in predicting emergency admission [20]. Most machine learning approaches show high accuracy (more than 70%) in prediction, which shows the great potential of applying machine learning techniques for analysing EHRs data and the importance of temporal data prediction.

Among the different features of EHRs data, temporal information in EHRs data is attractive because predicting patients’ future visits could help save hospital resources and insurance costs [21]. Zhang et al. [11] pointed out that the conventional knowledge representation of EHRs data always represents the visit record as a feature vector that abandons the temporal order. They proposed a bag-of-words matrix to highlight the temporal information and to use dynamic time warping algorithms for calculating temporal alignment between two sequences before similarity measuring. Xu et al. [22] further took both time-invariant and time-varying features from patients’ EHRs data to calculate the duration of patients in different care units (CU) by a modified point process model. Besides those techniques, the machine learning approach is often raised because some machine learning algorithms such as long short-term memory (LSTM) neural networks are designed for temporal sequence data. It seems that supervised learning is promising in dealing with EHRs data. Therefore, supervised learning will be the main direction for handling work injury cases in this project.

Lin et al. used the convolution neural network to extract hand position information from a single red-green-blue (RGB) video and then put temporal position data into an LSTM-based autoencoder to obtain low-dimension data for predicting bradykinesia, one of the essential features of Parkinson’s disease [23]. Their method aims to simplify the computer-aid motion processes learning by replacing traditional auxiliary smart garments with machine learning neural networks. Their method presents an idea of applying machine learning in aiding clinical diagnosis. However, they obtained few training samples and did not explain the machine learning model mechanism.

Lin et al. used a gated recurrent unit (GRU) in predicting intensive care unit (ICU) disease diagnosis with EHRs data [24]. They also tried to replace the traditional recurrent neural network (RNN) machine learning method with GRU, which is more potent in handling time-series data [25]. Later, they adopted GRU as an autoencoder trained by a generative adversarial network (GAN) model to encode the diagnosis and procedure information into an embedding vector for predicting drug–drug interactions [2]. The experiment is based on real-world clinical dataset MIMIC-III and yet did not involve any clinical experts in evaluating the result [26]. Since their work did not involve clinical experts examining their model and result, it might need more examination before being put into practical use.

Another common approach to handling the EHRs data is clustering before machine learning prediction [27]. Liu et al. [1] used a heterogeneous information network (HIN) to construct NetHealth data as a recommendation system to suggest individuals’ mental health states. They suggest that HIN can yield a genuinely multiplicative effect of data integration since time-series data are valued. Cheng et al. used health insurance data to construct a HIN and used the tensor decomposition method to cluster the patients before LSTM, predicting future hospital visits [21]. The purpose of clustering here is to minimize omission of patients’ information. The validation is based on previous data and is validated by clinical experts but does not involve any current data. Table 1 summaries the type of algorithms of the papers mentioned above.

Therefore, we propose a smart work injury management (SWIM) system to help the case managers to evaluate the situation of work injury and to estimate sick leave and rehabilitation plan. 

## 3. Methodology

Smart Work Injury Management (SWIM) System 1.0 is a prediction system concerning sick leave and the strategy for treating the injured worker [28]. The data generally contain two parts, namely, static and dynamic data. A work injury consultancy company has been collecting the data since 2000 in Hong Kong. We collected 90,154 work injury records (static data) containing the worker’s basic information and the incident. Inside these records, there are 15,515 work injury cases (dynamic data) which have logged the changes in the records. For example, dynamic data contain the historical change log of estimated sick leave (SL) and permanent disability (PD). The training machine learning model includes the final outcome of patients, duration of SL, percentage of PD, compensation cost, and result of legal dispute. Principal components analysis [29] and variational autoencoder (VAE) [30] are used to project the information from traditional and statistical forms to latent space form. Human factors of the case managers (self-experience) are also included and are turned into a rule-based system. Finally, a web-based application based on node.js and Microsoft Azure was built to predict the sick leave and strategy for treating new cases, as shown in Figure 1.

The idea of the machine learning process is simplified and explained in Figure 2. Once a new case is queried, the information of the injured worker and the incident (static data) will be encoded and projected into the latent space as Z_Start_. The sick leave progress (dynamic data) will be treated as Z_At_ and projected in the latent space. Then, both Z_Start_ and Z_At_ will be used to estimate the normal situation, of which Z_Normal_ will reflect the normal sick leave and treatment strategy. Severity cliff will also be found in the latent space and will estimate if the current case tends to be a high-level management case or not. Severity cliff will also be used to find the Z_HLM_, which will reflect the high level management (HLM) case’s sick leave as well as the strategy. Finally, K-nearest neighbours algorithm (KNN) [31] will be used to find cases similar to Z_Normal_ and Z_HLM_ in the latent space. Similar cases will be decoded. Estimated sick leave, permanent disability, treatment strategies, and related information of HLM and normal situation will be displayed to the user. 

### 3.1. Input Data

Input data are composed of static data and dynamic data. Static data include injured worker’s personal information such as age, gender, and salary. They also include work injury diagnosis information such as sick leave days, cause of injury, and injured body parts. Most of the data would only be recorded after a case is closed. Those data show injured workers’ entire situation and raise ideas about developing a neural network for machine learning prediction. Part three shows the static data distribution plot of each import column.

Dynamic data record the human manipulations of the system. Once the case manager receives the injured worker’s recent update, they will record those updates or modify the treatment strategy according to those updates. Dynamic data record those manipulations and the timestamp when it happens. Therefore, the dynamic data provide a good vision about what happens from the date of the accident to the close date.

#### 3.1.1. Static Data Information

Static data consist of 90,154 work injury records. Each record has 124 columns. The columns mainly fall into four categories: employee information, accident information, compensation and intervention, and IDs/Ref numbers. Out of the 124 columns, only 17 of them are used as training data of the neural network. Employee information includes gender, age, industry, etc. Accident information includes the date of the accident, injured body parts, cause of the accident, etc. The case manager’s intervention, such as HLM and alertness, are included. Alertness indicates whether the case manager thinks the case needs extra attention. Table 2 shows the input and output data of the model. We separate them into continuous and categorical data and preprocess the data before feeding them into the neural network.

#### 3.1.2. Dynamic Data Information

Dynamic data include change logs of 15,515 work injury cases. The strategy column’s change logs are the focus for training data, while change logs of some other columns are used for evaluation.

Case_ID, date, and type of strategy are extracted from the logs for training. Because of the lack of dynamic data, the strategy column from the static data is used to compensate for the missing records. Alongside the latent representation from the neural network output, the combined dataset is used to train a KNN classifier for strategy suggestion. The change log of columns, including estimated SL and estimated PD, are used for evaluation. By comparing model output with estimated sick leave and permanent disability % in different timestamps (relative to the starting date of each case), the KNN and neural network model’s performance in different timestamps during case development can be assessed.

Figure 3 shows the dynamic logs count for each strategy. It is seen that RTW—Full Duties has the most change logs. However, the number is relatively small compared to the whole static dataset, which has more than 90,000 cases.

To better understand the nature of data, we suggest running the PCA for identifying the key features and running the copula matrix approach for identifying the non-linear relationship [32]. 

#### 3.1.3. Data Preprocessing

The continuous data were processed to be a standard score. The categorical data were one-hot encoded to be a binary vector. The missing data were dropped because during the training process, we found that replacing the missing data with the mean value would pollute the prediction results. We split the preprocessed data into testing data and training data. The split ratio was 5%.

### 3.2. Latent Space and VAE

There is a need to reduce the cost of return-to-work management [33]. The goal of SWIM 1.0 is to utilize machine learning to predict the cost and development of a case in order to aid the case manager’s decision making. To achieve this target, we need to firstly model a case for computational purposes.

The key (conceptual) idea of SWIM’s AI module is to model a case as a dynamic path. Each injured employee (IE) will have their own path, and the path represents the history of his/her case. The start of a path represents the starting condition and details of a case (e.g., the IE is hit in the chest; he is male, 24 and works as a cargo mover), and the end of a path represents how the case ended (e.g., the IE returned to work normally after 30 days). The zigzag of the path represents how the case is influenced by events (e.g., the case manager has contacted the employee to check on his progress) as shown in Figure 4. Nag et al. [34,35] proposed a similar concept, but their approach was about personal health care. The details will be illustrated in the following paragraphs.

Within this conceptual model, cases that are similar will have similar paths. For example, most cases where the IEs suffer minor injuries will end within 10 days. The path of similar cases will act like case 0167 and case 9687, as shown in Figure 4. Similarly, cases that are different will have different paths (e.g., IE will have minor vs. major injury). For example, case 0167 and case 7231 behave in different ways, and their paths are different. Their differences can be seen where the paths start and end, including the shapes of the paths. Although cases may start at similar locations, this does not mean that they will progress or end similarly due to external (or unforeseeable) factors. 

We further specifically conceptualize serious and benign cases. The latter are considered cases that are “normal”. They are cases that require little intervention and can be expected to manifest in a common pattern (e.g., most employees will return to work within some timeframe without events worth noting). The former, however, requires special attention from case managers to keep the progress under control. These serious cases are ones for which we might expect very long recovering time, fraudulent behaviours, legal actions and other obstacles that prevent case closing. To separate serious and benign cases, we conceptualize a separate severe region. Case 7231 and case 8216 in Figure 4 are considered serious cases, while case 0167 and case 9687 are benign cases. The path of serious cases is expected to end somewhere in the severe region, such as case 7231, and more severe cases will end in legal regions, such as case 8216. Assuming that a newcoming case is a severe case, the path of a new case behaves like the blue path in Figure 4. Initially, it would end at a legal region without intervention. The dashed line indicates its original path. However, the conceptual model should be able to predict the border of the severe region, and the case manager could manage the case according to the model-predicted information and then change the path of the new case to behave like a benign case, as shown in the right-hand picture of Figure 4. 

We call this conceptual model the rose garden model. For the actual implementation, the case path aforementioned is modelled within a latent space, a hyperdimensional space constructed by a machine learning model [36]. We therefore refer to it as a latent path. A latent path consists of latent points where each represents a key event of the corresponding case at some point in time. To construct this latent space, a model based on a variational autoencoder (VAE) [30] is used. VAE is a generative machine learning model that can construct a meaningful abstract representation for data [37]. In our scenario, we use the static data to train the VAE model and to generate the latent space. We refer to case information that exists in the feature space as the information of a case is its feature.

Figure 5 shows the network architecture in this project. The data were fed into the encoder according to their data type. For the continuous data encoder, a one-layer linear fully connected (FC) network with the LeakyReLU activation function was used to project each continuous data item to a higher dimension. The categorical data encoder is a one-layer FC network with a LeakyReLU activation function. The encoded data were concatenated and fed into the processing layers. The processing layers contain three layers with the same number of neurons, which is 100. After the processing layers, the continuous data decoder and categorical data decoder decode the embeddings and generate outputs. Both the continuous data decoder and categorical data decoder have a two-layer FC with LeakyReLU as an activation function. After training the neural network, we project a case into the latent space. Therefore, a projected case becomes a latent case vector, a high-dimensional representation of a case. 

In SWIM1.0, input features include industry, position, physical demand level, age, salary, gender, injured body part, nature of injury, cause of injury, frequency of manual handling operation and alertness. The goal of SWIM is to predict the cost and development of a case. For SWIM1.0, these predictions are how long it takes for the IE to recover and return to work (SL), the permanent disability for the IE due to the injury (PD), whether the case requires high-level management (HLM) and a timeline reference which shows how similar cases progressed in their lifetimes. To predict them, two different but related methods are used, a neural network approach and a latent nearest neighbour approach. The neural network will predict the path of a new case while the KNN approach is responsible for finding similar cases around each path step in the latent space. In addition, the case features in SWIM1.0 are divided into either static or dynamic components. Static components refer to case data that will remain static. For example, the age/salary/job of an IE will remain the same throughout the case. The dynamic component, on the other hand, refers to data that will change with time. In SWIM1.0, the dynamic component includes the Day_Passed_, indicating how many days have passed since the case started, and HLM_Day_, indicating when and if HLM is applied. In addition, currently, the path of SWIM1.0 involves a starting and ending point, an event that the model considers to be HLM. 

The concept and the related interface of the system are summarized in Figure 6. We first discuss how latent paths actually manifest. Since feature F and latent space L are different domains but correspond to each other, it is necessary to map these two spaces. This can be achieved by utilizing an encoder E and decoder D. Given case feature f (f e F), it can be projected into L via z = E(f) where z is the latent representation of the case. With f = D(z), a latent representation can be projected to F for viewing. We can now see that a latent path Z can be formed by Z = {Z_0_, Z_1_, Z_2_}. When projecting a case c to L, we consider static C_s_ and the dynamic components. A case that has passed t days in L is therefore Z_t_ = E({C_s_, t}). Thus, cases start at z_0 = E({C_s_, 0}). 

The latent space is the computational representation of reality. To predict how a case will develop, we predict how a case will move in the latent space. This is related to [38,39,40] in manipulating/interpreting the latent space. Thus, the key to prediction is the predictor P which can predict where the case will end: Z_p_ = P(Z_t_). We can know that a case will end at day RTW_Days_ by decoding Z_p_ with D(Z_p_). Since cases are categorized as benign and serious cases, we assume that these cases are handled differently. If a case is benign, where the case will end will be predicted by Z_BLM_ = P_BLM_ (Z_0_). We only use the information already available at the start of the case, as it is expected that the case is benign and uneventful, meaning that the case has a common pattern that can be derived at the very beginning. If a case is severe, we assume that the case will involve HLM, and where the case will end is predicted by Z_HLM_ = P_HLM_ (Z_t_), where t is HLM_DAY. We can see that P_BLM_ and P_HLM_ are networks that predict different management behaviours (i.e., with and without HLM). Because PD is related to the nature of the case and how long it takes for the injured worker to recover, an estimator P_PD_ (Z_p_) can predict the PD, given that the case ends at Z_p_.

It is important to identify benign and serious cases. It is hypothesized that one important factor to determine is whether or not a case is benign or serious and for how long the case has progressed. Given an injury, it is reasonable to assume there is a common timeframe for an injured worker to recover. If the case is significantly longer than the said period, we can reasonably assume that such a case may be serious. Therefore, we introduce the HLM predictor P_HLM_ that classifies whether a case needs HLM or not. The classification value is L_HLM_ = P_HLM_ (Z_t_). As the number of days passed increases, it will become more and more likely for a case to require HLM, as it moves further and further away from its common timeframe. From here, we can see that P_HLM_ can provide a HLM cliff that separates benign and serious cases.

### 3.3. KNN

Based on previous research, a latent nearest neighbour approach is used for finding timelines for reference [41,42]. More than 45,000 past cases would be processed by the VAE model and would generate equal numbers of points with 100-dimension coordinates. When a new work injury case comes in, the model will process the case and generate a new 100-dimension coordinate. In this research, the K in the KNN method is set at 50 considering our computation time and limited cases. Since we are dealing with 90,154 case points, computing one new case will take around 10 s to find the nearest 50 samples. The computation time is acceptable at the current stage. If the case manager reports time-consuming issues, we will run the case indexing to improve the efficiency of KNN. According to the incoming case coordinate, the system will find the 50 nearest points. To find the nearest neighbour, the endpoint is used: Z_End_ = E(f, RTW_Days_). Thus, to find the reference timeline for a predicted case, we compare Z_P_ and Z_End_. 

Each of those 50 points would have a distance to the incoming case point. The system uses the distance to calculate the weights of each case. The system will then collect each case’s ground-truth data, such as their strategies and happening date after the date of accident (DOA). The system will calculate the probability of a strategy applied according to the distance-weighted average and its occurrence time in ground-truth data.

## 4. Result and Discussion

### 4.1. Result of the Prediction

The goal of SWIM 1.0 is to aid in the case manager’s decision making by providing an objective prediction of a case. Specifically, this is achieved by predicting the sick leave of the IE (SL), the permanent disability of the IE (PD) and whether the case requires high-level management (HLM). To evaluate SWIM 1.0′s ability in prediction SL and PD, a test set is created to compare its performance with that of a human. The test set has a total of 2932 samples. For each sample, a case manager has made an estimation on when the case will end at some point in time. The 2932 samples contain benign and serious cases. They are predicted separately. For the benign cases, the compared human prediction is the first prediction made. For the serious cases, the compared human prediction is the prediction nearest to the day when HLM is applied. We also showcase SWIM 1.0 performance difference when alertness is used or not used. When compared with human prediction, we assume alertness is used.

Table 3 shows the mean prediction errors compared to the ground-truth data. SWIM performed better when predicting SL for benign and serious cases and when predicting PD for serious cases. For SL of serious cases, human prediction has an average error of 154.857 days while SWIM1.0 has an average error of 107.447 days, which is a significant ~30% improvement. For benign cases, human error is 16.344 while model error is 9.737, an even more significant ~40% improvement. For PD, SWIM1.0 performed better only for serious cases. The human prediction error is 2.104 while SWIM is 1.329, yielding an improvement of ~37%. Although for benign cases, SWIM1.0 performed worse than humans, the difference is only ~7%. Figure 7 shows the error metrics of all industries in BLM and HLM conditions.

If the system predicts that the probability of a strategy that occurs is more than 40%, and that the ground-truth data have this strategy, then this prediction is considered as a correct prediction. The above figure shows the BLM and the HLM cases’ prediction accuracy separately. The reason for separating BLM and HLM cases is that the BLM case usually has much fewer strategies than the HLM case. BLM cases typically take less time than HLM cases and therefore have no need for extra treatment strategies.

### 4.2. Result of KNN

Each case would have a latent space coordinate after the machine learning model processes the case information. It is believed that similar cases have closer distances than the others. The K-nearest neighbours (KNN) method could help find similar target cases because of the small data quantity and poor dynamic data quality [42,43,44]. The system will then have a rough dynamic data prediction such as strategy and the date of strategy applied predictions.

More than 3000 cases have been prepared as test data to see if the KNN method can find similar cases and therefore predict the strategies. If the system predicts that the probability of a strategy that occurs is more than 40%, and the ground-truth data have this strategy, then this prediction is considered as a correct prediction. Figure 8 shows the BLM and the HLM cases’ prediction accuracy separately. The reason for separating BLM and HLM cases is that the BLM case usually has much fewer strategies than the HLM case. BLM cases typically take less time than HLM cases and therefore have no need for extra treatment strategies.

BLM cases usually have one or two strategies for each case. Therefore, in most cases, the accuracy is either 100% or 0%. From this test set result, 100% correct cases take the domain. However, in HLM cases, the accuracy drops down to 20–50%. More than half of the strategies fail to correctly predict. The less accurate scenario might be because of the lack of data. For example, some strategies show up less than 100 times in the whole training dataset and testing dataset. Therefore, the KNN method might not find those strategies because the training dataset is too small. In the future, we will use more oversampling techniques to solve the data shortage problem.

If we only look into some critical strategies such as “legal” and “DS”, we can see that the “legal” strategy has a prediction accuracy of 28.65% while the “DS” strategy has a prediction accuracy of 22.28%. The predicted “DS” strategy probability distribution is shown in the left-hand side of the Figure 9, and the predicted “legal” strategy probability distribution is shown in the right-hand side of the following figure. It can be seen that there is no prediction probability higher than 0.8 for both scenarios. We assume that this is because the VAE model cannot separate the HLM and the BLM cases. Therefore, the prediction probability is pulled down by those BLM cases because BLM cases usually have no “legal” or “DS” strategies.

## 5. Limitation and Further Improvement

Currently, SWIM 1.0 and case management of injured workers are one-way. This may be an ineffective and inefficient form of communication. The system predicts the path based mainly on static data. However, dynamic data could also contribute to the path prediction. Therefore, dynamically updating the model based on incoming dynamic data could be one of the future improvements. Dataset limitation is another concern in the project. The dataset only comes from one single case management company. Although the cases of the current company cover 15 kinds of industries and 523 corporations, the strategy of case management is affected by the policy of the case management company.

From the perspective of academia, there is also great value in improving return-to-work management [45]. The panel doctor and the injured worker will pass the physical document to the case manager. Then, the case manager will review the document and provide a suggestion. This is a passive and reactive approach. However, this planning may not be well rounded. This suggestion is also based on the experience of the case manager, which may be subjective. We will improve our work to SWIM 2.0 and enhance communication with different stakeholders and improve the RTW planning and RTW review with other cutting-edge technologies. Communication and collaboration between stakeholders are essential for RTW [46]. The development of this project consists of five technological insights as shown in Figure 10.

Enhance communication by APP and QR code technology. We will develop a user-friendly APP which involves QR code technology to replace document/email communication between the injured worker, case manager and panel doctor.Enhance communication through APP and OCRAI technology. We will embed the OCRAI technology to replace document exchange, data entry and verification between the injured worker and case manager.Enhance communication with smart contracts and blockchain technology. We will use the decentralized approach to handle the data storage and synchronization of a different isolated system among the stakeholders.Enhance job matching with artificial intelligence. We will improve SWIM 1.0 from a one-off prediction from the beginning stage to a continuous prediction and monitoring system. The medical data will enhance the latent space of SWIM 1.0 to estimate the job ability of injured workers in different stages. Various artificial intelligence types, such as rule-based, fuzzy measure and integral, will be used to develop a job-matching system, as shown in Figure 11.Enhance the job review with workflow management technology. We will adopt a paperless approach by using workflow management. All the documents will be changed into digital form and embedded into a well-defined structure in workflow management applications to enhance the job review process in RTW as shown in Figure 12.

## 6. Conclusions

In this paper, a smart work injury system was introduced. We came up with a conceptual model, namely the rose garden model, to describe the injured case path in a high-dimensional space. A NN model was trained to project the incoming case into the high-dimensional latent space, as the conceptual model describes. The incoming case is represented as a latent space vector. The vector was used later in the KNN model to search for the 50 most similar cases to find the possible strategies according to previous cases. In summary, we made three main contributions in this work: (1) We introduced a high-quality machine learning model to handle the multi-modality work injury data and to make a more accurate prediction than the human case managers. (2) We introduced a concept named the rose garden model to project a work injury case into a high-dimensional space where the case becomes a latent case vector, a high-dimensional representation of a case, which well represents the recovery path of an injured worker. (3) We developed a complete system that translates the machine learning result into human-readable results that could support modern work injury case management well. 

Our system is currently used by case managers. In the future, we will make further improvements and field tests after collecting feedback from case managers.

## Figures and Tables

**Figure 1 bioengineering-10-00172-f001:**
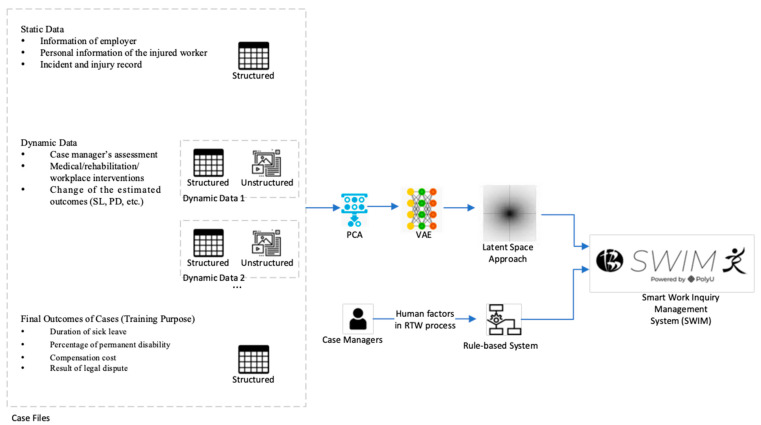
Overview of our current work (SWIM 1.0).

**Figure 2 bioengineering-10-00172-f002:**
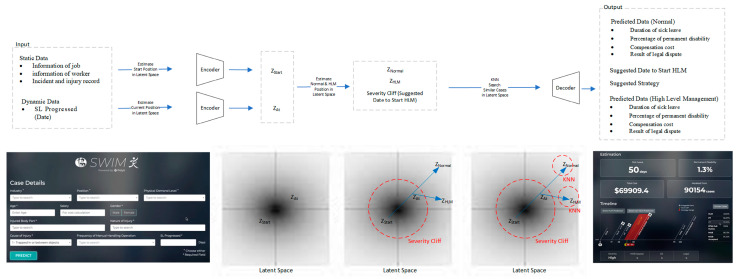
Overview of the machine learning and latent space concept in SWIM 1.0.

**Figure 3 bioengineering-10-00172-f003:**
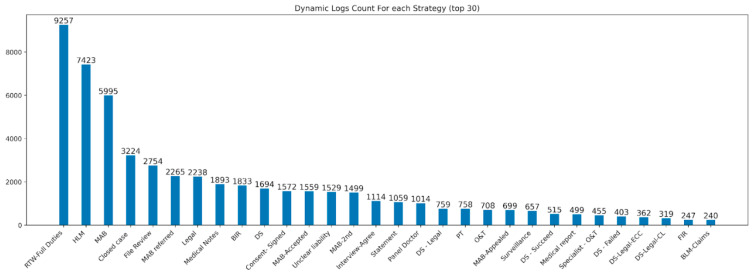
Dynamic logs count for each strategy.

**Figure 4 bioengineering-10-00172-f004:**
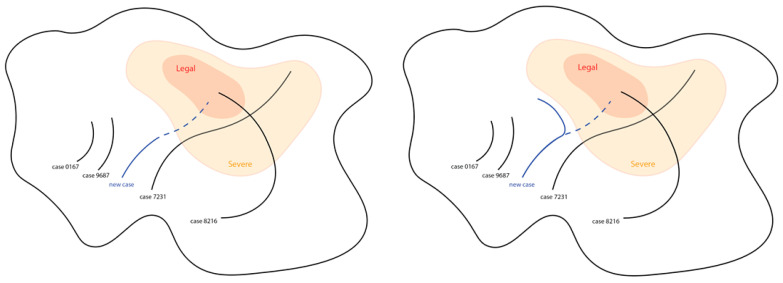
Concept of latent space in SWIM 1.0.

**Figure 5 bioengineering-10-00172-f005:**
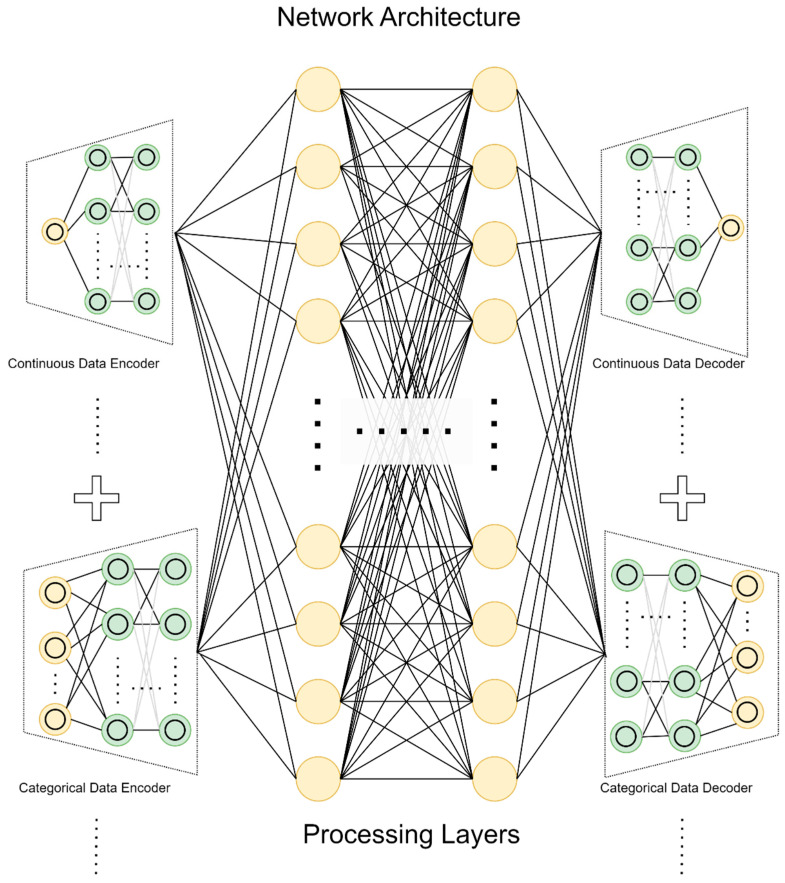
Network architecture.

**Figure 6 bioengineering-10-00172-f006:**
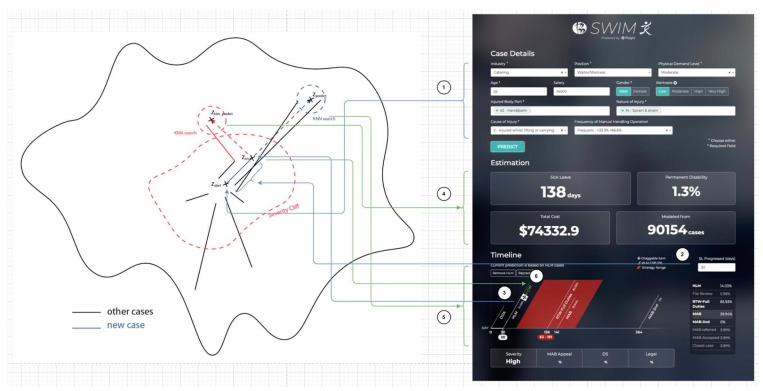
Sample screen of SWIM 1.0.

**Figure 7 bioengineering-10-00172-f007:**
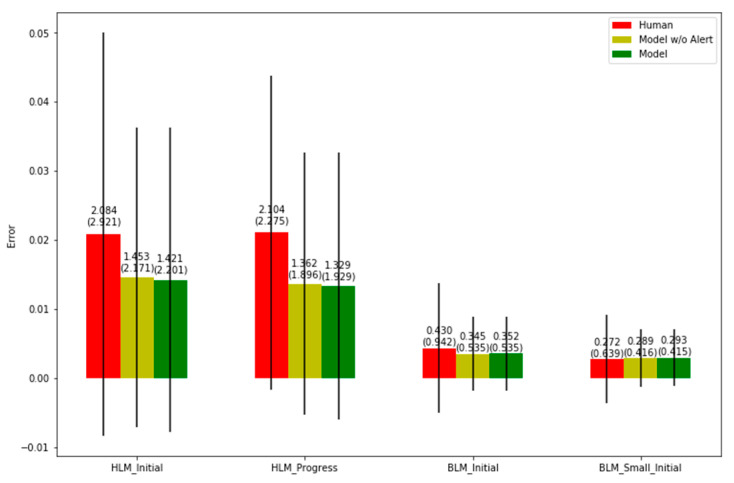
Error metrics of all industries (N = 2932).

**Figure 8 bioengineering-10-00172-f008:**
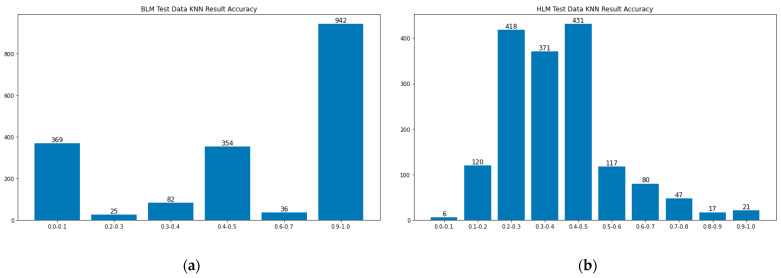
Accuracy of: (**a**) BLM test cases using KNN; (**b**) HLM test cases using KNN.

**Figure 9 bioengineering-10-00172-f009:**
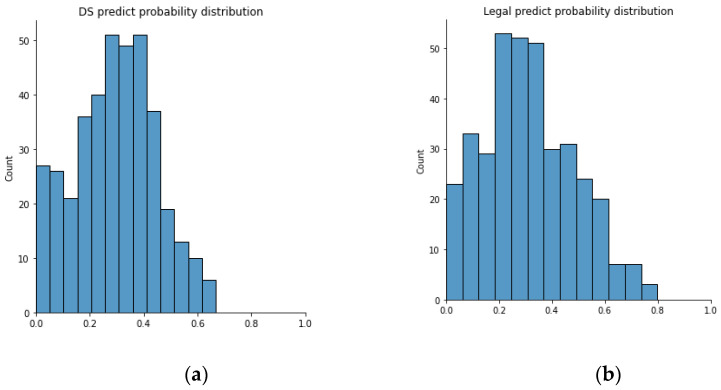
Distribution of: (**a**) predicted DS probability; (**b**) predicted legal probability.

**Figure 10 bioengineering-10-00172-f010:**
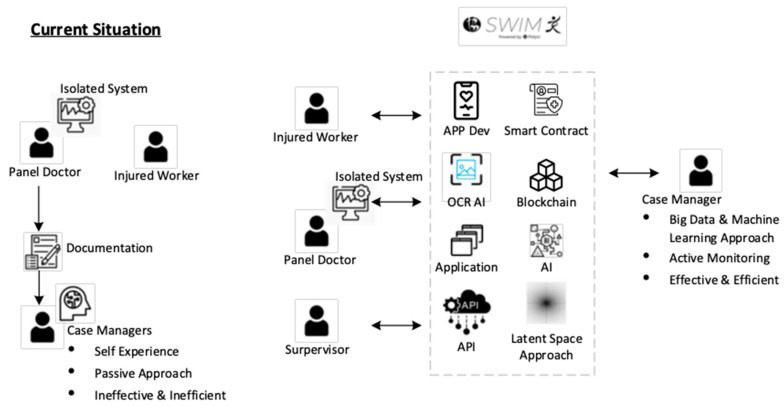
Overall concept of SWIM 2.0 development.

**Figure 11 bioengineering-10-00172-f011:**
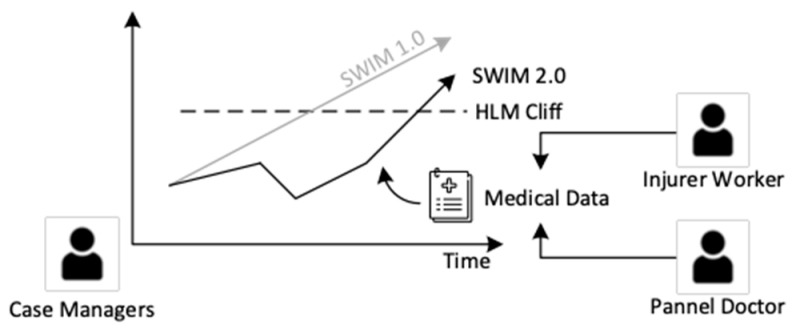
Comparison of SWIM 1.0 and 2.0 prediction model.

**Figure 12 bioengineering-10-00172-f012:**
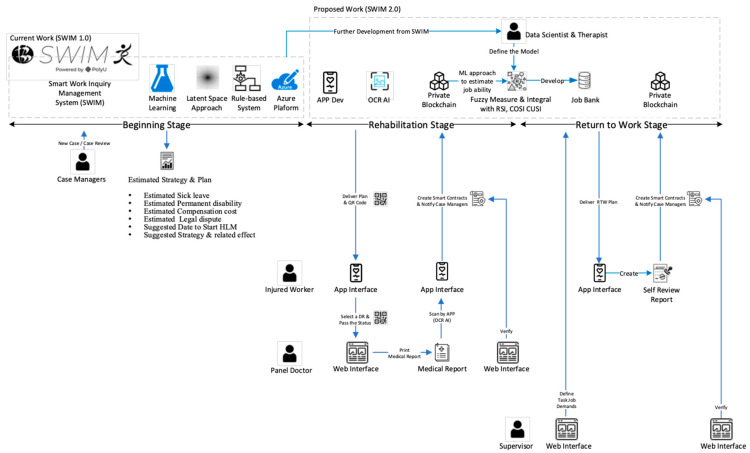
Overall flow of new case management in SWIM 2.0.

**Table 1 bioengineering-10-00172-t001:** Summary table.

Type of Algorithms	Name of Paper
Supervised Learning	Adversarially regularized medication recommendation model with multi-hop memory network [2] Deep learning for electronic health records: a comparative review of multiple deep neural architectures [5] Artificial intelligence analysis of EEG amplitude in intensive heart care [16] Utilizing electronic health records to predict multi-type major adverse cardiovascular events after acute coronary syndrome [17] DWE-Med: dynamic word embeddings for medical domain [18] Scalable and accurate deep learning with electronic health records [19] Predicting the risk of emergency admission with machine learning: development and validation using linked electronic health records [20] GGATB-LSTM: grouping and global attention-based time-aware bidirectional LSTM medical treatment Behaviour prediction [21] Bradykinesia recognition in Parkinson’s disease via single RGB video [23] DMMAM: deep multi-source multi-task attention model for intensive care unit diagnosis [24] Deep dynamic imputation of clinical time series for mortality prediction, information sciences [25]
Unsupervised Learning	Heterogeneous network approach to predict individuals’ mental health [1]
	A hierarchical fusion framework to integrate homogeneous and heterogeneous classifiers for medical decision making [27]

**Table 2 bioengineering-10-00172-t002:** Input and output data.

Data Category	Name of Fields	Data Type
Input	Age	
	Salary	
	Sick Leave Total Days (SL)	
	Days Until HLM	Continuous
	Alertness Score	
	Date of Accident (DOA)	
	Date of Case Closed	
	Injured Body Parts	
	Nature of Loss	
	Cause	
	Industry	
	Position	Categorical
	Position Physical Demand Level	
	Position Handling Frequency	
Output	Form7 PD (PD)	
	Sick Leave Total Days (SL)	Continuous
	Alertness Score	
	BLM/HLM	Categorical

**Table 3 bioengineering-10-00172-t003:** Results comparison.

	SL Prediction Error (Days)		PD Estimation Error (%)	
	BLM	HLM	BLM	HLM
Model w Alertness	9.727	107.447	0.293	1.329
Model w/o Alertness	9.663	121.373	0.289	1.362
Subjective (Human)	16.344	154.857	**0.272**	2.104

## Data Availability

Not applicable.

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
