# Peer review of "Smart Work Injury Management (SWIM) System: A Machine Learning Approach for the Prediction of Sick Leave and Rehabilitation Plan"

_bioengineering, 2023, doi:10.3390/bioengineering10020172_

Round 1
Reviewer 1 Report
Title:
Smart Work Injury Management (SWIM) system: A machine learning approach for the prediction of sick leave and rehabilitation plan
In this work, using a machine learning-based approach authors have enhanced the traditional prediction of disability duration and rehabilitation plans for work related injury and illness. They also have developed an interactive dashboard to visualize machine-learning
results.
In my opinion, the content of the manuscript (with some revision) is well-written and has a good result. Therefore, I suggest its publication in BIO_ENG, up to some changes indicated below:
1-Since you are describing the machine learning methods in the field of "Work Injury" and "Rehabilitation Case Management", it's better that you divide your literature review based on "supervised learning" and "unsupervised learning". Also, in the supervised learning section,
it's better that you discuss prediction/regression and classification separately.
2-what kind of preprocessing algorithms did you do in your work?
please mention them in section 3.
3- In both static and dynamic data, one of the most concerning will be the nonlinearity relationship between variables, so in your PCA algorithm maybe you need to implement this nonlinearity relationship.
It is better to check or mention such concern in section 3. or in figure 1 or in the conclusion. see, e.g., Sheikhi et al. (2022) below
4- please discuss in more detail your results and suggest the future path/extension of this work for readers in the conclusion.
------------------------------------------------------------
A Sheikhi, R Mesiar, M Holeňa (2022) A dimension reduction in neural network using copula matrix International Journal of General Systems
Author Response
Thank you for conducting a comprehensive and thorough review of our paper. We are most grateful for your comments on all requested aspects of the review and have the following responses:
(1-Since you are describing the machine learning methods in the field of "Work Injury" and "Rehabilitation Case Management", it's better that you divide your literature review based on "supervised learning" and "unsupervised learning". Also, in the supervised learning section,
it's better that you discuss prediction/regression and classification separately.)
We totally agree with your observation and comment. We have divided the literature review based on "supervised learning" and "unsupervised learning" and a table is added (Line 135) to summarize the literature review.
(2-what kind of preprocessing algorithms did you do in your work?
please mention them in section 3.)
Thanks for your suggestion. We have included a new session, 3.1.3 Data Preprocessing (Line 217).
(3- In both static and dynamic data, one of the most concerning will be the nonlinearity relationship between variables, so in your PCA algorithm maybe you need to implement this nonlinearity relationship. It is better to check or mention such concern in section 3. or in figure 1 or in the conclusion. see, e.g., Sheikhi et al. (2022) below)
Thanks for pointing this out. We have added the content and Sheikhi’s work in the Session 3.1.2 (Line 212)
(4- please discuss in more detail your results and suggest the future path/extension of this work for readers in the conclusion.)
Thanks for your suggestion. We have enhanced Session 5, Future Improvement (Line 425). Due to the ongoing development, we will continuously work on this topic, analyze the data and publish our work in Bioengineering.

Reviewer 2 Report
(1) The manuscript needs English proofreading.
(2) What is the limitation of the proposed approach?
Author Response
Thank you for conducting a comprehensive and thorough review of our paper. We are most grateful for your comments on all requested aspects of the review and have the following responses:
(The manuscript needs English proofreading.)
Thanks for your advice. We have hired a native writer to do the editing and proofreading.
(What is the limitation of the proposed approach?)
Thanks for pointing this out. We have included a new session, Session 5 Limitation and Further Improvement. (Line 424) to discuss the limitation of static data and the data source from one single case management company. Although the cases of the current company cover 15 kinds of industries and 523 corporations, the case management strategy is affected by the policy of the case management company.

Reviewer 3 Report
This paper describes a work injury management system which uses machine learning to predict sick leave and rehabilitation plans for injured workers. I consider this topic as a medical information system, which is inside of the scope of this journal. The topic is relevant and the problem to be solved is properly contextualized in the introduction. However, there are many gaps in methodology, which are detailed in the following review. Conclusions must be improved. Conclusions are discussing more about future works than about the contributions presented in this paper. Authors must clearly emphasize contributions and limitations of SWIM 1.0. There is no reference if this work was approved for some ethics committee. There is no discussion about the validation of this tool in a real environment. Detailed comments: L27. Avoid using keywords which already were in the title. You should use broader or more specific terms. Example: you can replace "machine learning" by "artificial intelligence"
L28. Wrong font size
L103. Unpresented acronym: RGB.
L110. Unpresented acronym: ICU.
L111. Unpresented acronym: RNN.
L133. "injured worker [33]. We"
L133. You should provide more detail about the dataset, such as collection place (Hospital, City, and Country) and period. L142. Unpresented acronym: SL, PD. L 147. The "." must be removed after Zat L151. ZHLM: HLM must be in subscript L173. You should provide a most detailed view regarding the dataset. You should cite, at least, the most important columns L175. You should cite the columns used as input to the NN. You should provide details about the training process, such as data split approach, missing values, normalization, binarized columns, and architecture. L209. Elements in figure 4 must be explained. What does the dashed line mean? What does the legal area mean? Case number does not have any meaning. You should exemplify severe and short injuries, normal and benign cases. You also should exemplify cases cited in line 215. Do you have a single model to predict each feature, or all predictions are generated by the same model? L224. You should explain better what means "they are not handled properly." L226. "The above mentioned is the rose garden model" It is not clear the meaning of the term 'above mentioned'. L233. "the data is case information (e.g., IE’s age/salary)" This statement is very imprecise. L241. How are the models combined: ensemble, bagging or sequential? You should provide more details. L247. Why are DAY_PASSED and HLM_DAY using this notation? L251-293. Notation must be completely reviewed. All uses of underscript font are incorrect. Probably the authors missed $ symbols delimiting formulas. L289. How was the K value defined? L302-314. This text is repeated (L285-299). However, it seems a good approach creating a subsection to explain each model: NN and KNN. KNN is memory intensive. How do you deal with this problem? L320. Missed point after (HLM) L327. Alertness concept was not explained. L329. Metrics are not explained. Did you use mean or median to values presented? L340. You should use the same names for table 1 and the legend of figure 6. In the same sense, terms in the x axis can be the same as table 1. L340. You should remove the term "all industries" from the title. L364. "For example, some Strategies show up less than 100 times in the whole training data set and testing data set." You should consider using some oversampling technique in this case. L380. Further improvement section is not useful in this paper. Probably, after implemented, these improvements will be a focus on other papers. Authors should use this space for emphasizing the contributions in the actual version of development. In this sense, there are some gaps which can be addressed, like detailing the NN and the combination of KNN and NN. L340. A typical way to evaluate prediction results generated by regression models is creating a scatterplot containing real values x predicted values, and compute the correlation coefficient. You should consider this kind of plot in section 4, to evaluate NN prediction results. L348. You should present the confusion matrix to discuss KNN classifier results. L442. Conclusions discuss more elements of future work than contributions presented in this paper. Authors should emphasize contributions and limitations of SWIM 1.0. There is no reference if this work was approved for some ethics committee. There is no discussion about the use of this tool in a real environment.
Author Response
Thank you for conducting a comprehensive and thorough review of our paper. We are most grateful for your comments on all requested aspects of the review and have the following responses:
(This paper describes a work injury management system which uses machine learning to predict sick leave and rehabilitation plans for injured workers. I consider this topic as a medical information system, which is inside of the scope of this journal. The topic is relevant and the problem to be solved is properly contextualized in the introduction.)
Thanks for your confirmation on it.
(However, there are many gaps in methodology, which are detailed in the following review. Conclusions must be improved. Conclusions are discussing more about future works than about the contributions presented in this paper. Authors must clearly emphasize contributions and limitations of SWIM 1.0.)
We totally agree with your observation and comment. We have modified the paper according to your advice. We have improved and added Section 5, Limitation and Further Improvement (Line 424) to emphasize contributions and limitations of SWIM 1.0. We have also enhanced the conclusion part by pointing out our contributions (Line 482)
(There is no reference if this work was approved for some ethics committee.)
Thanks for pointing this out. We agree that it is about mistake. We have added Session 8, Compliance with Ethical Standards (Line 495) to state our ethical approval process.
(There is no discussion about the validation of this tool in a real environment.)
Thanks for your suggestion. Our system is currently used by case managers. In the future we will have further improvement and field test after collecting a certain number of feedbacks from case managers. (Line 490).
(Detailed comments)
Thank again for conducting a comprehensive and thorough review. We have modified the paper and state the line number in the following comments.
(L27. Avoid using keywords which already were in the title. You should use broader or more specific terms. Example: you can replace "machine learning" by "artificial intelligence") Line 30.
(L28. Wrong font size) Line 31
(L103. Unpresented acronym: RGB.) Line 109
(L110. Unpresented acronym: ICU.) Line 116
(L111. Unpresented acronym: RNN.) Line 118
(L133. "injured worker [33]. We")(
L133. You should provide more detail about the dataset, such as collection place (Hospital, City, and Country) and period.) Line 136
(L142. Unpresented acronym: SL, PD.) Line 149
(L 147. The "." must be removed after Zat) Line 162
(L151. ZHLM: HLM must be in subscript) Line 166
(L173. You should provide a most detailed view regarding the dataset. You should cite, at least, the most important columns) Line 197
(L175. You should cite the columns used as input to the NN. You should provide details about the training process, such as data split approach, missing values, normalization, binarized columns, and architecture.) Line 186 – Line 233 Section 3.1.1 – 3.1.3
(L209. Elements in figure 4 must be explained. What does the dashed line mean? What does the legal area mean? Case number does not have any meaning. You should exemplify severe and short injuries, normal and benign cases. You also should exemplify cases cited in line 215) Line 240 – Line 273
(Do you have a single model to predict each feature, or all predictions are generated by the same model?) Line 274 – Line 287. We have added a new network architecture to explain the model.
(L224. You should explain better what means "they are not handled properly.") Line 260 – Line 263
(L226. "The above mentioned is the rose garden model" It is not clear the meaning of the term 'above mentioned'.) Line 265
(L233. "the data is case information (e.g., IE’s age/salary)" This statement is very imprecise. ) Line 272
(L241. How are the models combined: ensemble, bagging or sequential? You should provide more details.) Line 294- Line 298
(L247. Why are DAY_PASSED and HLM_DAY using this notation?) Line 302
(L251-293. Notation must be completely reviewed. All uses of underscript font are incorrect. Probably the authors missed $ symbols delimiting formulas.) Line 305 -Line 357 The notation has been rewritten.
(L289. How was the K value defined?) Line 345 – Line 346
(L302-314. This text is repeated (L285-299). However, it seems a good approach creating a subsection to explain each model: NN and KNN. KNN is memory intensive. How do you deal with this problem?) Line 340 – Line 357
(L320. Missed point after (HLM)) Line 363
(L327. Alertness concept was not explained.) Line 193
(L329. Metrics are not explained. Did you use mean or median to values presented?) Line 373 – Line 381
(L340. You should use the same names for table 1 and the legend of figure 6. In the same sense, terms in the x axis can be the same as table 1.
(L340. You should remove the term "all industries" from the title.) Line 382
(L364. "For example, some Strategies show up less than 100 times in the whole training data set and testing data set." You should consider using some oversampling technique in this case.) Line 412 – 413
(L380. Further improvement section is not useful in this paper. Probably, after implemented, these improvements will be a focus on other papers. Authors should use this space for emphasizing the contributions in the actual version of development. In this sense, there are some gaps which can be addressed, like detailing the NN and the combination of KNN and NN.) Line 377 – Line 495
(L340. A typical way to evaluate prediction results generated by regression models is creating a scatterplot containing real values x predicted values, and compute the correlation coefficient. You should consider this kind of plot in section 4, to evaluate NN prediction results. L348. You should present the confusion matrix to discuss KNN classifier results. As the scatter plotter and confusion matrix show the exact data (Sick Leave) of each case to the public, which may consist of the business value of the case management company and the privacy of their customers, we need to further discuss with the company. We apologize that we can only present the group data. We will discuss with the case management company and later offer more group data in different industries, job nature and injured part.
(L442. Conclusions discuss more elements of future work than contributions presented in this paper. Authors should emphasize contributions and limitations of SWIM 1.0. There is no reference if this work was approved for some ethics committee. There is no discussion about the use of this tool in a real environment.) Section 6 Ethics and Dissemination, Line 472 – Line 475 & Section 8 Compliance with Ethical Standards, Line 496 - Line 501 are added.
We are grateful that you were one of our reviewers and provided detailed comments to improve our paper. We really appreciate your time and contribution to this journal.

Reviewer 4 Report
-
Author Response
Thanks for your encouraging feedback. We will continuously work on this topic and analyze the data.